# Recent Progress in the Molecular Imaging of Nonalcoholic Fatty Liver Disease

**DOI:** 10.3390/ijms22147348

**Published:** 2021-07-08

**Authors:** Olivia Wegrzyniak, Maria Rosestedt, Olof Eriksson

**Affiliations:** 1Science for Life Laboratory, Department of Medicinal Chemistry, Uppsala University, SE-751 83 Uppsala, Sweden; olivia.wegrzyniak@ilk.uu.se (O.W.); maria.rosestedt@ilk.uu.se (M.R.); 2Antaros Medical AB, SE-431 83 Mölndal, Sweden

**Keywords:** positron emission tomography, imaging, immune cells, fibroblasts, collagen, fibrosis, NASH

## Abstract

Pathological fibrosis of the liver is a landmark feature in chronic liver diseases, including nonalcoholic fatty liver disease (NAFLD) and nonalcoholic steatohepatitis (NASH). Diagnosis and assessment of progress or treatment efficacy today requires biopsy of the liver, which is a challenge in, e.g., longitudinal interventional studies. Molecular imaging techniques such as positron emission tomography (PET) have the potential to enable minimally invasive assessment of liver fibrosis. This review will summarize and discuss the current status of the development of innovative imaging markers for processes relevant for fibrogenesis in liver, e.g., certain immune cells, activated fibroblasts, and collagen depositions.

## 1. Introduction

In normal physiology, fibrosis is an encapsulating and reparatory process in response to injury. However, excess formation and deposition of connective tissue, i.e., pathological formation of fibrosis, is an important feature in many diseases such as nonalcoholic steatohepatitis (NASH) [1]. Obesity and type 2 diabetes (T2D) are risk factors for developing NASH and nonalcoholic fatty liver disease (NAFLD). Upwards of 25% of the global population is estimated to currently suffer from NASH/NAFLD [2], and given the global epidemic of T2D [3], it is expected that patients with liver fibrosis will increase substantially during the coming years.

Persistent inflammation in the liver may trigger the activation of stellate cells and fibroblasts, which in turn are responsible for the production and deposition of the extracellular matrix such as collagen, which is a hallmark of NAFLD and NASH [4]. Furthermore, NASH must currently be verified and diagnosed through fine needle aspiration biopsy, which is invasive and potentially inaccurate depending on the heterogeneity of tissue [5]. Novel interventions aim to prevent, arrest or even reverse the progression of fibrosis by targeting a number of different pathways (or just weight loss) [6], but the drug development process is hampered by the lack of accurate and objective noninvasive assessments of change in fibrosis or fibrosis promoting processes [5].

Therefore, it is crucially important to develop noninvasive methods in order to detect, diagnose, stage, and study the molecular processes that drive the pathology of fibrosis. This would potentially contribute to an early-stage accurate staging of the disease and enable assessment of the effect of an intervention, such as lifestyle or drug treatment.

Clinical biomedical imaging techniques carry the potential of enabling minimally invasive assessment of disease. Modalities can broadly be divided into anatomical and molecular imaging, depending on the mode of recording. Clinical techniques with great potential for anatomical or structural imaging include, but are not limited to, computed tomography (CT), magnetic resonance imaging (MRI), and ultrasound (US). Molecular imaging (or functional imaging) involves the in vivo detection of molecular processes, enzymes, and receptors. Molecular imaging techniques in clinical, routine applications often involve contrast agents and include positron emission tomography (PET) and single-photon emission tomography (SPECT), as well as certain MRI sequences. In particular, PET is of interest in the setting of imaging of molecular processes involved in fibrosis, as it is a highly sensitive, quantitative, and non-observer-dependent medical imaging technique using radiolabeled molecules for tracking biological processes and receptors. Due to its high chemical resolution, PET can detect femtomolar changes in target concentration, making it ideal for detection and assessment of small changes in tissue concentration also of low concentration of receptors or cells [7].

The current review will discuss the development of novel contrast agents for the study of fibrosis in the liver. The focus will be on recent progress in PET radiopharmaceuticals, while also contrasting these developments with established imaging markers.

## 2. The Process of Pathological Fibrosis

### 2.1. Hepatic Steatosis

Among its many roles, the liver plays a key role in lipid metabolism [8]. Indeed, there are two sources of fatty acids. Either they are delivered to the liver through blood from lipolysis of triglyceride (TG) in adipose tissue, a process regulated by insulin on adipocytes, or they are synthesized from glucose and fructose by de novo lipogenesis (DNL) [9]. Then, lipids are mainly stored in hepatocytes as TGs, which is an inert and non-cytotoxic form of lipid. However, under pathological conditions such as in the context of non-NAFLDs, the disposal of fatty acids through beta-oxidation or the formation of TG can be overwhelmed. Therefore, these processes lead to the accumulation of TG in lipids droplets inside hepatocytes. This condition is called steatosis. NAFLDs can be subcategorized as NAFL (only evidence of hepatic steatosis on liver histology) or NASH (steatosis, lobular inflammation, and hepatocyte ballooning) [10]. However, recently, a panel of international experts has suggested another term: metabolic dysfunction-associated fatty liver disease (MAFLD). This new definition avoids the description “nonalcoholic” and is this time based on positive criteria. Indeed, the criteria to diagnose MAFLD are based on evidence of hepatic steatosis, in addition to at least one of the following three criteria: overweight/obesity, established type 2 diabetes, or evidence of metabolic dysregulation [11].

Lipid overload in hepatocytes can lead to the formation of lipotoxic species (toxic intermediates of triglyceride synthesis and their derivates), which results in endoplasmic reticulum stress, oxidant stress, and inflammasome activation. It has been shown that some immune cells can contribute to steatosis. Indeed, natural killer T (NKT) cells promote the uptake of lipids by hepatocytes [12]. The lipotoxicity is a driver of hepatotoxicity and induces apoptosis of hepatocytes. Consequently, damage-associated molecular patterns (DAMPs) are released in the microenvironment surrounding hepatocytes, which triggers an inflammatory reaction (Figure 1).

### 2.2. Immune Reaction and Fibrosis

DAMPs will activate hepatic stellar cells (HSCs) through toll-like receptors (TLRs) expressed at their surface [13]. Those cells are located in the space of Disse. Under physiologic conditions, they are quiescent, have a function of storage for retinyl esters, and contribute to liver regeneration. After activation through DAMPs, they transdifferentiate into myofibroblast-like cells and become proliferating and contractile [14]. They are responsible for fibrosis by abundantly producing extracellular matrix components (such as collagen I, II, IV). Activated HSCs (aHSCs) also induce inflammation through proinflammatory molecules, which reinforce the profibrogenic environment, and ensures the maintenance of HSCs activation [15]. These molecules can also be produced by different cells, such as HSCs, hepatocytes, liver sinusoidal endothelial cells (LSECs), and immune cells as neutrophils, macrophages, lymphocytes, NKTs, and natural killer (NK) cells. DAMPs also stimulate Kupffer cells (KC), which then produce cytokines to recruit inflammatory immune cells, as well as activate and promote the survival of HSCs. For instance, KCs produce interleukin 1 beta (IL1β) and tumor necrosis factor (TNF) that mediate the survival of activated HSCs [16]. IL1β also interacts with LSECs leading to the recruitment and the infiltration of a large number of neutrophils. In the micro-environment surrounding the neutrophils, cytokines, such as TNF are triggering the activation of neutrophils [17]. Neutrophils perform different functions, such as phagocytosis, granule release, reactive oxygen species (ROS) generation, and the formation of neutrophil extracellular traps (NETs) [18]. Activated KCs also release cytokines which leads to monocytic infiltrates. The recruited macrophages produce IL1β, TNF, transforming growth factor-beta (TGF-ß), and platelet-derived growth factor (PDGF) to induce the survival, activation, and proliferation of myofibroblasts.

LSECs also have an active role in fibrosis. First, after a hepatic injury, LSECs acquire a profibrotic phenotype and secrete ECM and profibrotic molecules. LSECs also regulate HSC activation through an alteration of the balance of vasodilators/vasoconstrictors molecules or by secreting TGF-β and PDGF. Then, LSECs interact directly with the immune cells and HSCs. Together with hepatocytes and possibly HSCs, LSECs can directly present antigens to T cells [19].

HSCs produce chemoattractant molecules to recruit lymphocytes. Myofibroblasts also promote lymphocyte migration through the secretion of cytokines, such as TGF-β. T-helper (Th) lymphocytes derived from CD4+ T cells play a key role in fibrosis. For instance, Th17 cells and regulatory T cells (Treg) are proinflammatory and produce cytokines that enhance liver inflammation and fibrosis. Indeed, they stimulate KC and macrophages to express inflammatory cytokines. It also stimulates HSCs to express collagen type I and enhances their differentiation into fibrogenic myofibroblasts [20]. Another example is Th2 cells that stimulate the production of immunoregulatory mediators in macrophages and profibrogenic gene expression in myofibroblasts. Then, NKT cells take part in steatosis by promoting the uptake of lipids by hepatocytes or by activating HSCs [21]. 

In addition, it has been shown that IL33 that is released from physically damaged hepatocytes is implicated in liver fibrosis. Indeed, it stimulates type 2 innate lymphoid cells (ILC2) to produce IL13, which in turn promotes the activation of HSCs [22].

In conclusion, different cells and molecules take part in fibrogenesis in the liver and thus represent markers of interest for molecular imaging.

## 3. Clinical Assessment of Liver Fibrosis

The clinical tests for diagnosis of NAFLD and NASH usually incorporate peripheral markers of liver damage such as ALT (alanine aminotransferase) and AST (aspartate aminotransferase). However, these and other proposed plasma markers are usually overlapping between stages of inflammation and fibrosis. Of course, sampling of peripheral markers obviously doesn’t discriminate between release from the tissue of interest and other organs. 

Imaging is often employed to directly study the changes in the liver. Clinically available imaging techniques, such as CT and MRI, are excellent for noninvasive visualization of lesions on the anatomical scale, i.e., lesions with increased density in the lung (especially CT), or even fat concentration in the liver (MRI). However, it is sometimes difficult to separate different similar processes, e.g., lung density increase can be due to fibrosis, but also edema and inflammation. in addition, the sensitivity is limited, meaning it may be a challenge to detect low-grade fibrosis (early in the disease) or small changes induced by interventions.

Increased fibrosis tends to make the liver less elastic and stiffer. Several recently developed techniques exploit this feature in assessing fibrosis grade. Transient elastography (e.g., FibroScan) uses ultrasound to evaluate liver stiffness. The examination is noninvasive and painless but somewhat observer-dependent. Transient elastography is especially sensitive in measuring a modest to a high degree of fibrosis, and can be used, e.g., to guide fine-needle aspiration biopsies for histopathological verification. Magnetic resonance elastography (MRE) is a special sequence which, just like ordinary MRI, is noninvasive. MRE is more accurate than transient elastography, and it has added the benefit of measuring the stiffness in the entire liver in an observer-independent manner [23,24].

NASH diagnosis currently requires biopsy verification by histopathology, by assessing, e.g., collagen content, fat content, inflammation, and hepatocyte damage (Figure 2) [25]. In many cases, the hepatic biopsy is acceptable, despite its association with non-negligible risks, e.g., bleeding or even life-threatening complications. However, repeated liver biopsy without direct benefit for the patient, e.g., outside of diagnosis that may directly influence treatment, is more difficult to motivate. This is especially true in interventional trials, where study design may dictate biopsies both at baseline and following treatment. In this setting, imaging of fibrosis or associated markers would potentially provide an important improvement in patient safety and enable larger-scale studies.

## 4. Molecular Imaging of Fibrosis—State-of-the-Art

### 4.1. PET as a Technique to Visualize Fibrotic Processes

PET uses sophisticated and rapid chemistry to label small molecules or proteins with positron-emitting radionuclides. The most common PET radionuclides have a radioactive half-life, ranging from minutes to days to match the ligand's biodistribution and excretion. For the vast number of applications, radionuclides with a half-life of less than 2 h are used (^18^F, ^68^Ga, and ^11^C). The selection of an optimal radioligand for in vivo imaging can be performed based on many different parameters, but perhaps the most important one is affinity. The sensitivity of a PET radioligand is usually expressed as the binding potential (BP) (Equation (1)). This equation says that for successful imaging of a certain target, either the protein density (B_max_) should be high or the affinity high (small K_d_, which means high affinity).
(1)$$\mathrm{BP}=\frac{B_{\max}\;\left(\mathrm{receptor}\;\mathrm{density}\right)}{K_{d}\;\left(\mathrm{affinity}\right)}\;$$

In the case of imaging of fibrosis in the liver, one may posit that the density of immune cells and fibroblasts involved in ongoing inflammation and fibrogenesis may be relatively low compared to the total liver volume. Thus, with a low B_max_, one should focus on PET radioligand with a very high affinity to ensure successful imaging of the intended target—often at least in the nanomolar range. 

For imaging of collagen deposition itself, i.e., the Collagen type 1 triple helix bundles, one may, on the other hand, suppose that the B_max_ instead is very high or almost unsaturable (as collagen type 1 bundles don’t have a receptor to imaging in the traditional sense). Thus, the affinity for ligands aiming to image collagen bundles may prove acceptable also in the micromolar range.

Finally, one may consider the aspect of radiation dosimetry. PET radioligand uses ionizing radiation for detection, which obviously becomes a limiting factor in many patient groups. In the oncological setting, the PET dose is almost negligible compared to the dose received from diagnostic scans or even radiotherapy. However, in the scanning for NAFLD and NASH setting, the patient populations are relatively healthy and the dose received from a dedicated PET scan should be limited for patient safety. Large protein ligands requiring long-lived radionuclides (e.g., ^89^Zr, ^64^Cu, ^124^I) may generally yield higher radiation doses, and dosimetry limitations must be considered. This similarly holds for short liver radionuclides, but the effective dose is generally low for this category. The dosimetry aspects are especially important in the drug development aspect since repeated exposures (before and after treatment intervention) are desired. 

Another potential drawback of the PET technique is related to its cost and availability. PET scanning requires substantial infrastructure, including GMP qualified radiochemistry labs and often cyclotron for on-site production of radionuclides. Thus, PET scans are relatively costly compared to, e.g., CT scanning. Additionally, PET scanning capabilities are usually only present at major institutions, limiting its use in routine diagnostic workup at smaller hospitals and in primary care. Transport of PET tracers from central radiochemistry hubs to local sites are expected to improve this situation in the future, as it the increasing reliance of radiometals such as ^68^Ga, which can be produced by generator rather than cyclotron.

### 4.2. Imaging the Immune Context in Liver

First, some surface proteins, which are specifically and strongly expressed, could be suitable as imaging targets for the visualization of activated immune cells in the liver. Thus, specific binding of radiotracers to immune cell receptors could be exploited for molecular imaging and might provide insight into immune responses.

The chemokine receptor 2 (CCR2) is expressed on monocytes and macrophages, among other cells. Previous studies have shown that CCR2+ macrophages accumulate in periportal areas in patients with NASH and advanced fibrosis. Besides, CCR2 inhibitors are effective in reducing fibrosis in animal models of NASH and fibrosis by targeting disease-promoting liver macrophages. This makes CCR2 inhibitors a potential treatment for fibrotic NASH [26]. Therefore, the development of a radioligand allowing the analysis of CCR2 expression in a noninvasive approach would be of interest for the staging of fibrosis as well as for the development of new therapies, such as CCR2 inhibitors. Recently, the ^64^Cu-DOTA-ECL1i PET radiotracer has been characterized in disease-related lung inflammation in an LPS mouse model as well as in human lung tissue, including patients with chronic obstructive pulmonary disease (COPD). The radiotracer is based on a 7 amino acid peptide, the extracellular loop 1 inverso (ECL1i), which binds to CCR2. The PET results of this study showed that the signal obtained was specific as it was significantly higher in the lung, the liver, and bone marrow of the LPS group compared to the control group. In addition, the signal was significantly lower in blocking studies using non-radiolabeled ECL1i in excess or in CCR2-deficient mice. Moreover, ^64^Cu-DOTA-ECL1i was stable in mouse serum after 1 h. In humans, autoradiography (ARG) studies of lung tissue sections from COPD and donor subjects show that the binding of ^64^Cu-DOTA-ECL1i is proportional to the numbers of CCR2-positive cells [27]. Recently, the Food and Drug Administration (FDA) approved the use of ^64^Cu-DOTA-ECL1i for clinical trials. Thus, multiple clinical trials for lung inflammation, head and neck cancer, or carotid atherosclerosis are currently ongoing at the Washington University School of Medicine [28]. However, the use of ^64^Cu-DOTA-ECL1i also has limitations. Indeed, it has a very rapid blood clearance, which could limit the sensitivity of detection. Furthermore, the use of ^64^Cu, which has a decay half-life of approximately 13 h, may be less adapted for imaging for expanded clinical use. Despite its limitations, this tracer, which is currently not tested in the context of liver disease, may be interesting to be applied in the context of hepatic fibrosis thanks to its promising characteristics to see if it can be effective.

Another immune receptor that is of interest for the diagnosis and the monitoring of the patients is the interleukin-2 receptor (IL2-R). During the inflammation, T cells secrete the interleukin-2 (IL2), which binds to the IL2-R. It is formed by three subunits: IL2-Rα (CD25), IL2-Rβ (CD122), and IL2-Rγ (CD132). It is expressed on activated cytotoxic T cells or also regulatory T cells. CD25 is particularly interesting as it is an already known marker of T-cell activation. Previous studies show that the level of expression of CD25 is associated with advanced fibrosis NAFLD [29]. Interleukin-2–Derived Radiotracers for PET Imaging of T Cells have already been evaluated in mice. The N-(4-**^18^**F-fluorobenzoyl)-interleukin-2 (^18^F-FB-IL2), which is currently used in a clinical trial in metastatic melanoma at the University Medical Center, Groningen [30]. However, this radiotracer has a major limitation, its complex and time-consuming production. More recently, the aluminum 18F-fluoride-(restrained complexing agent)-IL2 (^18^F-AlF-RESCA-IL2) and ^68^Ga-gallium-(1,4,7-triazacyclononane-4,7-diacetic acid-1-glutaric acid)-IL2 (^68^Ga-Ga-NODAGA-IL2) were developed, both with a simpler and faster production. They show good stability in serum after 1 h. In vitro binding assays were performed on activated human peripheral blood mononuclear cells (hPBMCs), which contained cells with more CD3-positive T cells expressing CD25 than the nonactivated hPBMCs. The uptake of the radiotracers was significantly higher in the activated hPBMCs compared to the nonactivated hPBMCs. Especially, ^18^F-AlF-RESCA-IL2 whose uptake was considerably higher than that of ^68^Ga-Ga-NODAGA-IL2 and ^18^F-FB-IL2. In addition, in vivo assays were performed in severe combined immunodeficient (SCID) mice inoculated with activated hPBMCs. ^18^F-AlF-RESCA-IL2 again showed higher uptake in PBMCs and lymphoid organs such as the spleen, bone marrow, and lymph nodes compared to the other radiotracers [31]. Thus, ^18^F-AlF-RESCA-IL2 that shows promising characteristics in mice model or ^18^F-FB-IL2, currently in clinical trials, could be used in future studies for the detection of CD25-positive immune cells in liver diseases.

Then, in addition to radiotracers targeting receptors, it is also interesting to use tracers capable of displaying a specific enzymatic activity revealing the immune context. For instance, myeloperoxidase (MPO) is a target of interest in NASH. An increase in MPO is implicated in the pathogenesis of NASH. This enzyme is released in the setting of neutrophil degranulation and proinflammatory M1 macrophages and it is associated with oxidative stress. It may play a key role in the development of liver fibrosis through the activation of HSCs [32]. Currently, several MRI and SPECT imaging studies to image MPO have already been conducted. The gadolinium contrast agent MPO-Gd, for example, was used to detect MPO activity in a NASH mouse models (ethionine and choline–deficient (MCD) diet) and human specimens. This experiment shows that MR Imaging of MPO allows distinguishing of NASH noninvasively from steatosis in NAFLD [33]. Recently, an ^18^F-labeled PET tracer to target MPO activity was developed, namely the 18F-MAPP. It consists of an activatable MPO activity radioprobe that binds to proteins and accumulates at site activity. PET imaging using ^18^F-MAPP was performed on complete Freund’s adjuvant (CFA) paw inflammation mouse model. The results show that the uptake on the CFA-injected side was four times higher than that on the PBS-injected side. Moreover, the radioprobe shows promising qualities for in vivo imaging, such as a short blood half-life, stability in plasma, and it does not demonstrate cytotoxicity [34]. What makes ^18^F-MAPP a promising translational candidate to noninvasively monitor MPO activity and inflammation in patients, and potentially, knowing MPO’s role in NASH, to monitor NASH pathogenesis. 

Finally, soluble markers are promising biomarkers for PET imaging of NASH. Although, PET imaging of soluble markers is generally challenging due to the potential background signal from the target protein in the blood. However, these cytokines are often found at very high local concentrations in tissue, which likely could generate a sufficiently strong PET imaging signal.

Tumor necrosis factor (TNF) is a proinflammatory cytokine secreted from activated macrophages and involved in many aspects of the regulation of the immune response, including the survival of HSCs, which are associated with enhanced liver fibrosis [35]. Recently, Beckford-Vera et al. developed a novel PET radiotracer for imaging of TNF in transgenic human TNF-expressing mice model of rheumatoid arthritis (RA). This radiotracer is based on a TNF inhibitor, namely the certolizumab pegol (CZP); it is a PEGylated humanized Fab fragment anti-TNF approved for the treatment of different immune diseases such as rheumatoid arthritis and Crohn disease [36]. It was modified with p-isothiocyanatobenzyl-deferoxamine (DFO) and radiolabeled with ^89^Zr. In vitro assays showed that [^89^Zr]DFO-CZP binds human TNF with very high affinity which allows the detection of lower levels of TNF expression. PET/CT Studies in transgenic mice overexpressing TNF showed that [^89^Zr]DFO-CZP uptake was significantly increased in hind paws and joints of transgenic RA mice compared to the control group. Moreover, the results from the immunostaining of TNF correlated with the uptake of [^89^Zr]DFO-CZP observed in the ex vivo biodistribution study supporting the specificity of the tracer for TNF. Furthermore, despite the large size of the molecules (90 kDa), it has a good tissue penetration for its size. Indeed, the uptake in ankle joints was seen after 4 h post-injection, which is faster than expected for macromolecules of this size. Thus, this study demonstrated that it is feasible to image hTNF in vivo with this novel radiotracer enabling better patient selection for treatment with anti-TNF therapeutics and also a following of the evolution of the pathology in the patient [37]. Nevertheless, the tissue penetration is still poor compared to smaller molecules. Moreover, in the latter stages of the disease, it became more difficult to differentiate the signal in the shoulders from the surroundings. This could be explained by the presence of TNF in tissues near these joints and by the resolution of the scanner. With knowledge of the importance of TNF in the NASH, [^89^Zr]DFO-CZP is a promising tracer to monitor TNF in NASH.

Similar to TNF, interferon-gamma (IFNγ) is a proinflammatory cytokine predominantly produced by activated Type 1 T helper (Th1) and Cytotoxic CD8+ T cells. NASH is characterized by excessive Th1derived IFNγ, and the proportions of IFNγ+ and TNF+ circulating CD8 T cells are also enhanced [38,39]. Thus developing a radiotracer for molecular imaging would give an insight into the immune context and help in the diagnosis or prognosis of NASH. A monoclonal antibody (Ab) PET tracer targeting IFNγ was developed. It is based on anti-IFNγ, rat IgG1 Ab called AN-18, labeled to ^89^Zr using desferrioxamine. PET imaging of BALB/c mice treated with CpG-ODN to stimulate IFNγ, showed higher uptake in the spleen of CpG-ODN–treated mice compared with untreated controls 72 h post-injection of the radiotracer. Moreover, uptake in tissues responsible for excretion like the liver and kidneys was low. Then, the specificity was confirmed by a competitive binding experiment with Ab in excess, in which the spleen uptake was significantly lower. In the same experiment, another PET study was performed on neu+ TUBO tumor-bearing BALB/c mice. When those mice receive HER2/neu DNA vaccination, the uptake in tumors is significantly higher than in nontreated mice. This also means that intratumoral IFNγ levels are low without treatment. Thus, ^89^Zr-anti-IFNγ tracer uptake can be indicative of response to therapy in cancer vaccination [40]. However, as the radiotracer is based on a rat Ab, it could not be used clinically as it may lead to unwanted immunogenicity in humans. Another limitation would be that T cells are densely present in physiologic secondary lymphoid tissues, such as the thymus, spleen, and lymph nodes, which may limit tumor-specific T-cell imaging. However, in this study, using a mouse model, it is the first to investigate the utility of an immunoPET probe for detecting IFNγ; it also shows the possibility of carrying out molecular imaging via PET of a soluble molecule as a cytokine.

Another proinflammatory cytokine that plays a major role in NAFLD is the interleukin 1 beta (IL1β). Neutrophils, monocytes, macrophages, and dendritic cells mainly secrete it. It has been demonstrated that it promotes liver steatosis, inflammation, and fibrosis. Indeed, IL1β has a key role in innate immune responses as it induces the production of profibrogenic cytokines. It also affects insulin signaling and stimulates triglycerides and cholesterol accumulation [41,42]. In addition, IL1β stimulates HSCs activation, proliferation, and survival [42]. A hamster mAb anti-IL1β was labeled to ^89^Zr (^89^Zr-α-IL1β), and PET imaging was realized to detect inflammation in murine colitis. The experiment was conducted on a dextran sodium sulfate (DSS) colitis mice model, in which an increase in colonic IL1β concentration was observed relative to control mice. PET imaging and ex vivo analysis revealed that in DSS colitic mice, distal colonic uptake of ^89^Zr-a-IL-1b was increased compared to control mice and correlated with colitis severity. Therefore, this experiment demonstrates that immuno-PET of innate immune mediators, such as ^89^Zr-α-IL1β, has a strong potential for diagnosing and monitoring inflammatory bowel disease [43]. Further, a similar experiment could thus be conducted on the NAFLD model. 

Nevertheless, those experiments were realized using such large molecules as antibodies (150 kDa), which reduced tissue penetration and excretion that may limit their use for imaging. Promising alternatives to antibodies could be used, such as nanobodies, aptamers, or affibodies.

To conclude, innovative radiotracers allowing imaging of the immune context by targeting cells, receptors, enzymatic activity or cytokines, have recently been developed. Those radiotracers are particularly interesting and could potentially help in an early diagnostic, staging and in the development of new drugs. 

### 4.3. Targeting of Fibroblasts or Stellate Cells

Another possibility in order to display fibrosis in NAFLD is to directly image fibroblast or stellate cells that play a pivotal role in fibrogenesis. 

The translocator protein (TSPO), for instance, is a biomarker that can be targeted in molecular imaging for monitoring the progression of hepatic fibrosis to cirrhosis. Indeed, TSPO is mainly expressed on mitochondrial membranes in HSCs and macrophages. TSPO plays a role in the promotion of the transport of cholesterol across mitochondrial membranes [44]. TSPO gene expression increases along with profibrotic genes in the liver during fibrogenesis. The *N*-benzyl-*N*-methyl-2-[7,8-dihydro-7-(2-[^18^F]uoroethyl)-8-oxo-2-phenyl- 9H-purin-9-yl]-acetamide ([^18^F]FEDAC), is a radioligand that specifically targets TSPO with high affinity (Ki, 1.34 nM). PET scans were realized on rats after 2, 4, 6, and 8 weeks of CCl4 treatment as a model of liver fibrosis versus control rats. Results from the PET scans showed that the liver tracer SUVs are significantly higher in all treatment groups, compared to controls. Moreover, uptake of the radiotracer correlated with TSPO expression and with CCl4 duration, as well as the severity of liver damage. Therefore, [^18^F]FEDAC could represent a useful imaging biomarker for noninvasive monitoring of liver fibrosis [45].

Platelet-derived growth factor receptor b (PDGFRβ) is a known biomarker of aHSCs. Its expression level is significantly increased on aHSC. While its expression is not detected, quiescent hepatic stellate cells (qHSCs). In liver fibrosis, overexpression of PDGF by aHSCs leads to several behavior changes of HSCs in the process of activation, including proliferation, migration towards chemokines, and loss of retinoid droplets [46]. An overexpression and excessive signaling has also been detected in cancers. Thus, a study evaluated the efficacy of an affibody molecule to monitor the expression of PDGFRβ in glioblastoma xenografts. For this reason, in vivo studies using this radiotracer were performed on mice implanted with a U-87 MG glioma cell line. The results showed that ^111^In-DOTA-Z09591 has a strong affinity for PDGFRβ, as it shows a K_D_ for binding to U-87 MG cells of 92 ± 10 pM. In vivo, its uptake was PDGFRβ -specific and showed a high contrast of the PDGFRb-expressing xenograft shortly after injection (3 h). This means that 111In-DOTA-Z09591 could be labeled to short-lived positron emitters [47]. As this radiotracer was able to visualize the expression of PDGFRβ in glioblastoma xenografts in a mice model, it would be promising in the context of NASH to study activated hepatic stellate cells in the liver.

Integrins are also promising targets in imaging liver fibrosis. For instance, αvβ1, αvβ3, αvβ5, αvβ6, and αvβ8 are interesting as they can release active TGFβ, a robust fibrogenic signal marker in the liver. Integrins transmembrane receptors that recognize the sequence R-G-D in its ligands. Thus, RGD peptidomimetic small molecules have been developed as integrin-antagonists and may provide new treatments for liver fibrosis [48]. For instance, CWHM12, a potent inhibitor of αV integrins, was used on a choline-deficient, L-amino acid-defined, high-fat mouse model of NASH. The results from this experiment demonstrated that CWHM12 was efficient in decreasing already established fibrosis and reducing hepatocyte apoptosis [49]. A recent study target was integrin αvβ3, as it could provide a noninvasive method to track hepatic fibrosis progression. The vitronectin receptor integrin αvβ3 triggers fibrogenic activation of HSCs. Moreover, it has been demonstrated that the more fibrosis progress, the more the protein of integrin αvβ3 will be expressed in human tissue and it is mostly expressed on activated HSCs. In vivo studies were held on CCl4 and bile duct ligation [BDL] mice. Basically, in this study, a [^18^F]-alfatide, which is a dimer of RGD peptides, was shown to have high affinity and specificity towards integrin αvβ3. Both animal liver fibrotic models and human liver tissue data suggest that [^18^F]-alfatide can detect fibrosis progression [50].

Another integrin of interest in liver diseases is the αvβ8 integrin. Indeed, immunohistochemical analysis of liver biopsies of children with liver fibrosis due to biliary atresia has shown that αvβ8 integrin is overexpressed compared with healthy controls [51]. Moreover, it has been demonstrated that the depletion of hepatocyte αvβ8 integrin leads to an increase in hepatocyte proliferation and acceleration of liver regeneration after partial hepatectomy in mice [52]. Therefore, in vivo mapping of αvβ8-integrin expression might enhance the prognosis of liver disease. For the moment, a novel radioligand, namely the ^68^Ga-triveoctin, has been developed to track αvβ8-integrin and has been tested in the context of melanoma. PET scans with the use of the cyclic RGD octapeptide trimer ^68^Ga-triveoctin, was realized on MeWo (human melanoma) xenografted SCID mice and one healthy human. From the results obtained, it can be conjecture that this new tracer is promising to map αvβ8-integrin in conditions associated with TGF-β dysregulation. As the tracer has a high affinity for the integrin, shows sensitive in vivo imaging in a murine model and a favorable biodistribution with low background in human [53].

Those novel radiopharmaceuticals based on RGD for in vivo mapping of integrins expression might enhance the diagnosis, prognosis of liver fibrosis, as well as support TGF-β-targeted therapeutic development.

Then, the fibroblast activation protein (FAP) is a cell-surface serine protease only expressed by aHSCs, but not by quiescent HSCs. Moreover, it is mainly expressed in regions with active fibrogenesis. Thus, FAP is now considered to be a potential biomarker to display liver fibrosis and as a promising therapeutic target [54]. A recent study has shown promising results in the visualization of cancer-associated fibroblasts (CAF) with the use of the quinoline-based PET tracers, which act as FAP inhibitors (FAPI), namely the ^68^Ga-FAPI. Indeed, the fibroblast activation protein (FAP) is overexpressed by the CAF in head and neck cancers (HNCs). In vivo, PET-CT was realized on patients with HNCs. It led to high-contrast images with ^68^Ga-FAPI owing to very specific and high tracer uptake in the tumor [55]. Therefore, ^68^Ga-FAPI would be a promising radioprobe to noninvasively monitoring the evolution of liver fibrosis. Currently, clinical trials are held at Fujian Medical University in order to evaluate the potential value of ^68^Ga-FAPI-04 PET/CT for the diagnosis and prognosis of liver fibrosis disease [56].

The endocannabinoid system plays a pivotal role in acute and chronic liver injury. Within it, two specific G-protein receptors have been identified that may play a role in liver disease, namely, cannabinoid receptor 1 (CB1) and cannabinoid receptor 2 (CB2). CB1 and CB2 are expressed in all liver cell types at different basal levels. For instance, CB1 has been detected in hepatocytes and endothelial cells. Both receptors are upregulated during chronic liver damage and mediate different functions. Although the impact of CB2 is controversial, it is hypothesized that the CB1 receptor activity upregulation brings about an increase of de novo fatty acid synthesis and a decrease in fatty acid oxidation, as well as an increase of lipogenic gene expression, and a decrease of secretion of triglyceride-rich very-low-density lipoprotein [57]. Thus, overexpression of CB1 is a hallmark of hepatic fibrosis as it promotes fibrogenesis. Several radiotracers have been developed for imaging CB1 receptors involving [^11^C]OMAR, [^18^F]MK-9470, [^18^F]FMPEP-d2, or [^11^C]SD5024, which are current radiotracers for Human PET Imaging, mainly used in the context of neuropsychiatric disorders [58]. Among these, [^18^F]MK-9470 was tested in hepatic encephalopathy (HE), which is a severe neuropsychiatric complication of both acute and chronic liver failure. This radioligand used in PET imaging of mice with bile duct ligation (BDL)-induced HE, has shown to be selective, and has a high affinity for the CB1 receptor both in the brain and liver. Therefore, [^18^F]MK-9470 seems to be a promising radioligand for noninvasive diagnosis and staging of HE [59]. Going even further, knowing the role of CB1 in the fibrogenesis in NAFLD and the existence of promising and seemingly effective radioligand in the liver, [^18^F]MK-9470 could be potentially interesting in the diagnostic or prognostic of liver fibrosis. 

Aquaporins (AQPs) are a family of transmembrane channel proteins expressed in all tissues. Among this family, aquaglyceroporins (AQP3, AQP7, AQP9, and AQP10) are permeable to water and small solutes. The liver expresses the four types of aquaglyceroporins, with AQP9 being the most expressed. Moreover, AQP9 expression is higher on the hepatocyte membrane compared to the other few tissues where it is expressed. Also, AQP9 is the main pathway for glycerol import from portal blood to hepatocytes. Thus, a change in AQPs in liver fibrosis could reduce the glycerol transport into hepatocytes. It has also been shown that AQP9 expression is reduced after the activation of HSCs in the advanced liver fibrosis stages [60]. PET/CT imaging in vivo with aminoglycerol (AR) labeled to 11C ([^11^C]AR) was performed on rats with thioacetamide (TAA)-induced liver fibrosis. Results from in vivo studies showed that the accumulation of [^11^C]AR in rat liver correlates with liver glycerol metabolism dysfunction during fibrosis progression. Uptake in the liver was significantly different among different groups of liver fibrosis stages. Thus, [^11^C]AR is a promising probe for noninvasive diagnosis and staging of liver fibrosis [61]. However, ^11^C could be less convenient for clinical use because of its short half-life (20 min).

To conclude, imaging fibrosis can be done by using radiotracers that target biomarkers expressed by fibroblast or stellate cells. They can be biomarkers that are usually not expressed on nonactivated cells, increased on activated cells, or even decreased in the context of fibrosis. Imaging of activated myofibroblasts would potentially provide a biomarker for the production of new collagen, i.e., fibrogenesis. Radiolabeled FAPI analogs are currently under clinical evaluation, and their sensitivity for imaging of tissue fibroblasts may soon be available. 

### 4.4. Targeting of Collagen Deposits 

Collagen is an important structural component in the extracellular matrix and it plays a crucial role in tissue development, which includes a balance of production and degradation. A disruption of this balance could cause overproduction and excess of collagen, leading to accumulation of collagen in different organs such as the liver, causing hepatic fibrosis and later leading to conditions such as NASH. Molecular targeting of collagen in combination with clinical imaging techniques such as MRI and PET could contribute to the development of noninvasive methods for detecting fibrosis at an early stage. The promising aspect of molecular targeting and imaging of collagen is that involvement of the molecular pathways in fibrogenesis includes several fibrotic disease processes [62], meaning that imaging agents shown to be suitable for assessing one fibrotic disease could have high potential to assess others.

Fuchs and colleagues developed an MRI contrast agent using a collagen-specific imaging peptide, EP-3533 [63]. The peptide, consisting of 16 amino acids, having an affinity of 1.8 µM, was initially implemented in a study for the identification of myocardial scars [64]. Functionalizing the probe for MRI through the addition of GdDTPA chelators, the group was able to demonstrate that the EP-3533 agent was able to distinguish between different stages of liver fibrosis in a CCl_4_ mouse model. MRI images showing the stomach, liver, and muscle of a control mouse and an Ishak score 5 mouse demonstrated higher uptake of EP-3533 in the liver of the fibrotic animal in comparison to the control mouse. These results were strongly correlated to the changes in the liver:muscle contrast upon the injection of EP-3533; it was demonstrated that the ratio increased with increasing Ishak score (i.e., disease progression) in the fibrotic CCl_4_-treated mice. 

Due to the perceived higher sensitivity of PET compared to MRI, the EP-3533 peptide scaffold was radiolabeled by ^68^Ga via a DOTA chelator. The resulting construct, called ^68^Ga-CBP8, demonstrated promising results in the imaging of preclinical models of lung fibrosis [65]. Thus, PET imaging using the EP-3533 peptide would be promising as it would allow to image collagen accumulation in liver fibrosis. 

Finally, PET imaging using ^68^Ga-CBP8 was tested on healthy volunteers and in subjects with idiopathic pulmonary fibrosis (IPF). Results show that in addition to being safe and well-tolerated, the ^68^Ga-CBP8 signal in lungs is significantly increased in IPF subjects compared to healthy volunteers 1-h after the injection. Moreover, the signal correlated with fibrotic lung regions determined by CT. It was also seen in regions determine as normal on CT, implying that the tracer may detect active collagen deposition that is not yet visible by CT [66].

Then, collagelin additionally targets collagen type III. Collagelin was identified from a screen focused on the platelet glycoprotein VI, which is the main platelet receptor for collagens I and III. In vivo imaging using ^99^mTc-collagelin was performed in mouse models of pulmonary fibrosis and myocardial infarction. The signal from the tracer correlated with areas of histologically confirmed fibrosis. Moreover, the use of this tracer in scintigraphic molecular imaging provided a contrasted signal between the fibrotic scars and healthy tissues [67]. Later dosimetric studies using two radiotracers based on collagelin analogs, namely [^68^Ga]Ga-NO2A-Col and [^68^Ga]Ga-NODAGA-Col were realized on healthy rats. In vitro experiments showed that those radiotracers bind specifically to histologically-confirmed fibrotic areas in cryosections of dog left ventricle myocardium. In addition, the tracers showed a quick clearance from blood and normal tissue after intravenous administration to healthy rats [68].

ProCA32.collagen1 was presented by Salarian and colleagues as a collagen 1 targeting contrast agent for the detection of liver fibrosis and NASH [69]. This protein-based agent was designed by attaching a collagen type 1 targeting peptide moiety to the contrast agent ProCA32 (creating ProCA32.collagen1), with the affinity of 1.42 µM to collagen1, determined by ELISA-method. In two animal models, TAA/alcohol model and NASH diet model, it was demonstrated that 24 h post-injection of ProCA32.collagen1 the uptake increased in fibrotic livers with early and late stages of fibrosis. 

Despite these promising results, both CBP8, collagelin, and ProCA32.collagen1 exhibit a relatively poor affinity for collagen, in the range of 1.4–2 µM, and thus relatively large amounts of binding sites must be present at the lesion, and high concentrations of radiotracer must probably administer to allow detection. Normally, PET tracers must have an affinity in the nanomolar range to reach sufficient sensitivity to detect their target.

Another promising radiotracer targeting type I collagen for the molecular imaging of fibrosis was described. It was evaluated in rat models of lungs, as well as liver fibrosis model. A fragment sequence from pro-MMP-2, a precursor of an enzyme interacting with type I collagen, was labeled with ^99m^Tc. SPECT/CT imaging with ^99m^Tc-CBP1495 was performed on bleomycin-induced pulmonary fibrosis and CCl4-induced liver fibrosis rat models. CBP1495 was shown to be effectively bound to collagen type I with an affinity of 861 nM. The in vivo imaging demonstrated that the uptake of ^99m^Tc-CBP1495is significantly increased in fibrotic lung or liver compared with healthy rats. In addition, CBP1495 has been shown to be an efficient probe to visualize collagen in vitro, ex vivo, and in vivo. However, the probe still needs to be optimized as its circulation time is too short, limiting its accumulation in target tissues [70]. Nevertheless, the results from this experiment demonstrated that molecular imaging using a radiotracer target collagen leads to higher uptake in the fibrotic liver compared with healthy rats. Thus, this confirms the promising potential of the PET radiotracer cited previously. 

Finally, Federico et al. developed a series of small peptides targeting Collagen type I fibrils based on engineering of the Decorin protein structure [71]. One of these peptides, LRELHLNNN, exhibited an affinity of K_d_ = 170 nM to Collagen type I, which is almost a 10 times improvement compared to the previously described collagen targeting peptides above. The LRELHLNNN scaffold was radiolabeled with ^18^F or ^68^Ga via chelator functionalization and showed promise by in vitro and in vivo PET [72]. However, the scaffold must be optimized concerning in vivo stability for further progress.

To conclude, innovative radiotracers directly targeting collagen deposits were developed as they may be useful, for instance, in detecting fibrosis at an early stage. Most of them were already validated on animal models, and ^68^Ga-CBP8 was tested on humans with lung fibrosis. More work has to be done to bring these tracers to clinical use on MAFLD patients, and they will then be of great help in the diagnostic and prognostic, as well as a predictive tool.

### 4.5. Targeting of Hepatocytes in the Liver

Hepatocyte stress and cell damage may constitute an important source of cytokine signaling for immune cell activation at the onset of inflammation. For example, injured hepatocytes have been described as a major source of IL33, which activates Type 2 Innate Lymphoid cells—which in turn activate HSC via the IL13 pathway. Additionally, hepatocytes are increasingly displaced by increasing amounts of fat and collagen in the liver as NAFLD progress to NASH. 

Thus, imaging of hepatocyte density and function may constitute an indirect assessment of liver fibrosis. There exist several hepatocyte contrast agents in the recent literature. The most established is probably the gadolinium-based MRI contrast agent gadoxetate. Gadoxetate MRI has been proposed as a marker for viable hepatocyte content and is undergoing preclinical [73] and clinical [74] validation for this purpose. A potential drawback with this technique is the reluctance to use Gd-based contrast agents in patients with renal insufficiency, potentially limiting its use in populations with a high incidence of diabetes. 

A PET tracer was recently developed for the hepatocyte-specific asialoglycoprotein receptor (ASGPR), which is known to become decreased as NASH progresses [75]. Liver activity and hepatocyte function could be tracked longitudinally as diet and CCl4 induced liver fibrosis development. 

Another clinically available PET radioligand is ^68^Ga-Tuna-2, which targets the glucagon receptor [76]. The glucagon receptor is, of course, the receptor of the endogenous glucagon peptide hormone and is expressed specifically on the surface of hepatocytes where they assist in the metabolic homeostasis of glucose by contributing to, e.g., glycogenolysis upon activation. ^68^Ga-Tuna-2 binding in the human liver has so far only been evaluated in patients with well-controlled T2D, and thus, its uptake in individuals with moderate to severe hepatic fibrosis is unknown [77]. 

## 5. Summary and Conclusions

As reviewed and summarized above, there is an increasing amount of available radiopharmaceuticals targeting many processes of interest in the cascade ending in fibrogenesis in the liver. There are PET markers in early clinical phases (and several in the preclinical phase) targeting surface markers on hepatocytes, immune cells, and activated HSC, in addition to DAMPS, cytokines, and collagen deposits. As of now, the most clinically advanced PET tracers for fibroblasts and collagen deposits are ^68^Ga-FAPI analogs and ^68^Ga-CBP8, respectively. Both of these tracers are currently being studied in proof-of-concept clinical trials, and the outcome of those will be crucial for future study design when considering the translation of other PET tracers for fibrosis described here. 68Ga-FAPI analogs, in particular, are mainly investigated for the imaging of cancer-associated fibroblasts, where the presence of FAP on fibroblasts is a marker for poor prognosis—as FAP is in fact a fibrolytic protein that assists in remodeling the tumor microenvironment allowing tumor growth and progression. Thus, it will be interesting to further understand if FAP expression in, e.g., liver fibroblasts constitute a marker for poor prognosis or a sign of fibrolysis and fibrosis resolution. With such specific imaging techniques emerging for clinical use, it will potentially be possible to detect a quantifiable signal arising specifically from certain cell types and activation states. By detecting a specific cell type, one may infer the number of binding sites (e.g., receptors) and even cell density in the liver compartments, in different stages of the disease, or in response to different interventions. With PET, such detection is feasible in vivo, in living subjects, while it earlier only was possible in biopsies by, e.g., microscopy techniques. 

Initially, it is most likely that PET techniques for imaging fibrosis-related processes will be mainly applied in research studies, e.g., to assess the effect of an intervention, rather than in clinical diagnosis or fibrosis staging. This is mainly related to the high cost of PET imaging, the limited availability of PET scanning capacity at primary care centers, as well as the associated radiation burden from the administered PET tracer. However, with time and increased understanding of the performance of the discussed PET tracers, one may foresee a broader application of particularly sensitive probes. For example, if one could demonstrate high sensitivity and specificity, e.g., staging of different degrees of liver fibrosis, it may turn out cost-effective to include PET scanning in the diagnostic workup in certain patient populations—similar to how PET is used today in oncology. Thus, in conclusion, emerging molecular imaging techniques may change the way we diagnose, treat and understand liver fibrosis in the coming years.

## Figures and Tables

**Figure 1 ijms-22-07348-f001:**
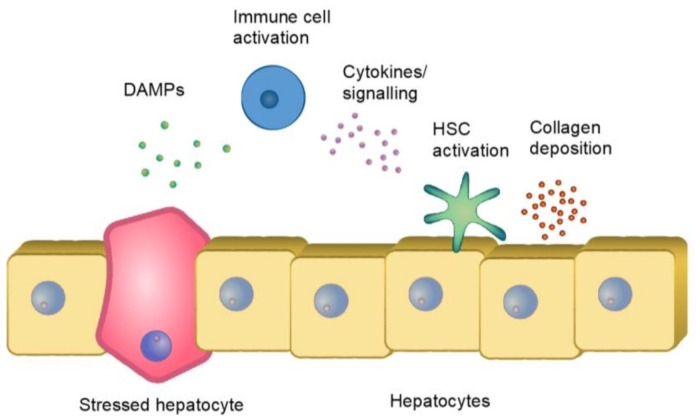
Schematic overview of a potential fibrotic cascade in occurring in the liver sinusoid microenvironment, from hepatocyte injury, via immune cell activation and signaling, to increased production of extracellular matrix (collagens) from hepatic stellate cells.

**Figure 2 ijms-22-07348-f002:**
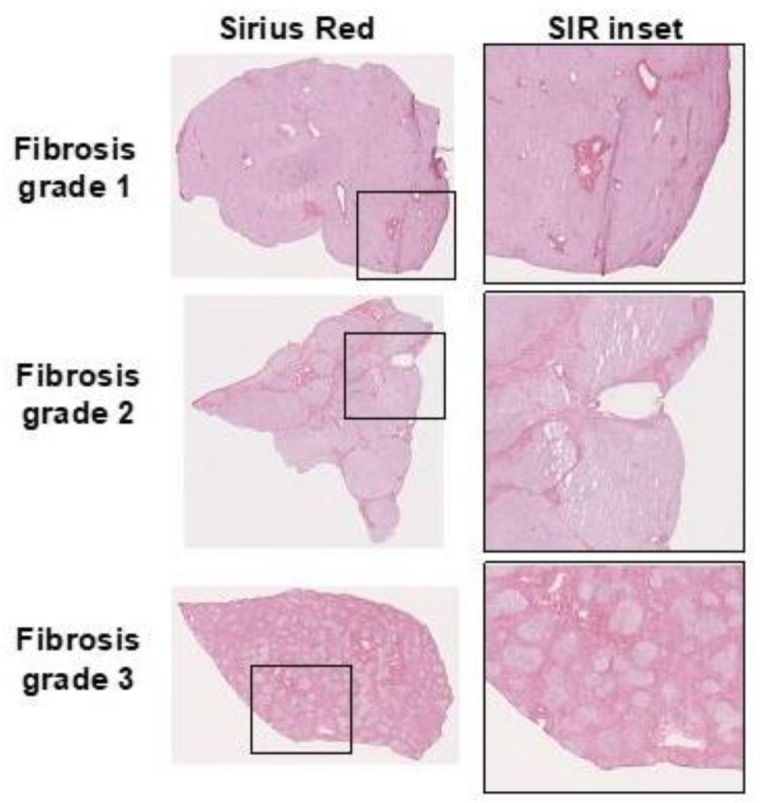
Example of collagen deposits demonstrated by Sirius red (SIR) staining in human liver sections with increasing fibrosis degree.

## Data Availability

Not applicable.

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
