# Peer review of "Recent Progress in the Molecular Imaging of Nonalcoholic Fatty Liver Disease"

_ijms, 2021, doi:10.3390/ijms22147348_

Round 1
Reviewer 1 Report
see attachement

Author Response
Recent progress in the molecular imaging of liver fibrosis 2 3 Olivia Wegrzyniak1 , Maria Rosestedt1 , Olof Eriksson1,2*
Authors general comments: We would like to thank the reviewers and editors for taking their time to read and comment on the current manuscript. We have tried to follow the reviewers suggestions and hope the manuscript is now acceptable for publication. Figure 3 has been removed as we have applied for reuse at the publisher but not yet received a response. Additionally, this figure is not crucial for the paper and it therefore seems proper to remove it.
REVIEW
1)TOPIC – „hot“, as NAFLD poses huge and increasing burden to all branches of medicine / hepatology; it is becomming the No.1 cause of chronic liver diseases, liver transplant etc. Directed pharmacotherapy is only in the pipeline and clinical trials are being hindered by the need of repeated liver biopsies. Therefore, non-invasie, dynamic tests of fibrosis (discerning delta on therapy) are unmet need.
Authors response: Thanks, we agree.
2)English: I recommend one more spell – check: the density of small clerical errors is disturbing. Otherwise English is acceptable.
Authors response: OK, this has been corrected.
3)Readability: Thank to celar writing, text itself is easily understandable. In the context, however, the very core topic is not allocated required weight (as compared to general comments on other diagnoses and clinical contexts) and is overshadowed by non-liver informations.
Authors response: In general we agree with the comment, but unfortunately most immune and fibrosis PET techniques have been evaluated in a wide range of disease models. However, also PET tracers evaluated in other models may be a clear interest in NAFLD/ NASH. If all those techniques were removed, and only those directly evaluated in animal liver disease models, we feel the review would give a limited overview of the field. The other tracers mentioned could potentially be tested in animal liver disease models or in some cases even human scanning in NASH. Thus, the broad scope could give readers more ideas – especially readers outside of the PET field.
We have tried to minimize the text around non-liver models (≈one sentence per tracer) to not be superfluous, but still give the reader a context.
4)Digestability for less informed readers: as concerns what in 2.1. (lines 96-143) does not directly relate to the imaging, shorten this section substantially: do not mention details concerning all the molecules which are not directly involved in imaging of the liver; only name them – e.g. „various molecules such as ....“; otherwise see above – 3)
Authors response: We have removed some of the excessive detail.
5)TEXT PROPER: - remove the sentence on the lines 32+. Fibrosis in NASH is not painful even in the advanced / terminal stages of cirrhosis: Fibrosis development in liver is 32 not intrinsically painful at early stages and may therefore avoid diagnosis until the disease 33 has become severe, including development of liver cirrhosis.
Authors response: OK, this has been corrected.
- CHAPTER 3. ASSESSMENT OF LIVER FIBROSIS
o provide mean sensitivity and specificity figures for TE and MRI and their negative and positive predictive values for the presence / absence of significant fibrosis.
Authors response: Very important comment. We considered adding this information, but realized we could introduce a bias in which values we add (and which study we cite). Several studies are published eg on TE, but the sensitivity and specificity is crucially dependent on the patient population (NASH grade, cirrhosis). In cirrhosis both sensitivity and specificity of TE is 80-90% for example. This is the subject for a review all on its own!
Furthermore, since specificity and sensitivity in the clinic is unavailable for all the discussed PET tracers (some exciting studies are on their way though), we suggest to keep this open and not single out a subset of publications.
o Insert very short paragraph on laboratory tests of fibrosis (FIB-4, APRI, ...) an their collective sensitivity / specificity
Authors response: We agree this is of utmost importance as plasma markers are routinely used in ongoing fibrosis interventional trials. However, plasma markers are usually taken as exploratory markers, and the exact selection varies considerably between trials. We have used eg cytokeratin 18 and similar in the past. More detail and specific numbers on sensitivity and specificity would require adding a broader context to not introduce a bias in publication citing, and we would like the current review to stay focused on the PET techniques.
o Finish the chapter by sentence wrapping it up: that, taken together, abovementioned diagnostic modalities and their combinations notwithstanding, there is still an unmet need for a new noninvasive tests for fibrosis. To bridge to - /introduce the next chapter
Authors response: A brief wrap-up summary has been added to each chapter, as well as in Chapter 5.
- Chapter 4: 4.2. The domains are well covered to understand the context but the whole text is undigestable for reader inspired by the headline: „“... liver fibrosis“; shorten / decrease granularity / be more generic in - the detailed informations about the non-liver use of various tracers. Explicitly state at the end of each section if there currently is or is not any use of a particular method in liver diseases – experimental or human. ; and state your opinion about its feasibility therein The absence of studies in liver disease will not decrease the value of paper and might stimulate researchers to re-direct their aims.
Authors response: Please see the answer to 3) above – same topic. We feel narrowing down the tracer to only those evaluated in liver diseases model, could lead to underreporting of promising targets of tracers tested in eg pulmonary fibrosis models. The lung fibrosis model is often “easier” to induce and scan, therefore sometiems selected over NASH models in animal imaging studies. But the tracers may be just as interesting for imaging eg activated myofibroblasts in liver, as in lung!
- Chapter 5: introduce the chapter with wrap-up sentence: how far from clinical use this bunch of techniques really is: e.g. is any of them ready for the prime time, or in foreseable future, or „more work has to be done“. Which of the generic domains under 4.2.- 4.5. seem/s currently most promising; and inside the domains, which items do you consider to be the most promising?
Authors response: We have added summaries of the techniques closest to human trials (and the most promising according to us)
o Add your outlook to the future: what is needed and what is to be expected? Are we talking about research diagnostic modalities or do you see them in a real-life clinical practice?
Authors response: We have added a discussion on this to Chapter 5.
o mention the cost and availability now and in the future
Authors response: This has been added to the PET chapter 4.1
Reviewer 2 Report
Wegrzyniak O et al. reported the recent progress in the molecular imaging of liver fibrosis. This review summarized and discuss the current status of development of imaging markers for processes relevant for fibrogenesis in hepatic certain immune cells, activated fibroblasts and collagen depositions. In general, this topic addressed is interesting and deserves a constructive discussion. The following is my comments to the authors.
- Title: In this article, the authors focus on molecular imaging in Non-alcocholic Fatty Liver Disease (NAFLD). So, the authors should include NAFLD [not Non-Alcoholic SteatoHepatitis (NASH)] in the title.
- Abstract. The first sentence is incorrect. Pathological fibrosis of the liver is a landmark feature in all chronic liver diseases, including NAFLD/NASH.
- page 4. MRE. The outcome is similar to Transient Elastography→This sentence is incorrect. The authors cite the following two papers, which should be discussed carefully.1,2
- page 4. NASH diagnosis currently requires biopsy verification by histopathology, by assessing e.g. collagen content, fat content, inflammation and hepatocyte damage (Figure 2). This reviewer believes that the authors did not go into detail regarding the pathological diagnosis of liver biopsy for NAFLD. Importantly, immunostaining for PDGFRB is not common; Azan staining and Masson-trichrome staining are common. Please cite the important review on pathological diagnosis of NAFLD.3
- The authors discuss advances in imaging on a cell-by-cell basis, but they should discuss which cell imaging is closest to clinical application when categorized into (1) the cellular level, (2) animal models, and (3) in humans.
References
- Imajo K, Kessoku T, Honda Y, et al. Magnetic Resonance Imaging More Accurately Classifies Steatosis and Fibrosis in Patients With Nonalcoholic Fatty Liver Disease Than Transient Elastography. Gastroenterology. 2016;150(3):626-637 e627.
- Park CC, Nguyen P, Hernandez C, et al. Magnetic Resonance Elastography vs Transient Elastography in Detection of Fibrosis and Noninvasive Measurement of Steatosis in Patients With Biopsy-Proven Nonalcoholic Fatty Liver Disease. Gastroenterology. 2017;152(3):598-607 e592.
- Brunt EM, Kleiner DE, Carpenter DH, et al. NAFLD: Reporting Histologic Findings in Clinical Practice. Hepatology. 2021;73(5):2028-2038.
Author Response
1.Title: In this article, the authors focus on molecular imaging in Non-alcocholic Fatty Liver Disease (NAFLD). So, the authors should include NAFLD [not Non-Alcoholic SteatoHepatitis (NASH)] in the title.
Authors response: OK, this has been corrected.
2. Abstract. The first sentence is incorrect. Pathological fibrosis of the liver is a landmark feature in all chronic liver diseases, including NAFLD/NASH.
Authors response: OK, this has been corrected.
3. page 4. MRE. The outcome is similar to Transient Elastography→This sentence is incorrect. The authors cite the following two papers, which should be discussed carefully.1,2
Authors response: The references have been added. We considered adding more information on TE and MRE eg sensitivity, but realized we could introduce a bias in which values we add (and which study we cite). Several studies are published eg on TE, but the sensitivity and specificity is crucially dependent on the patient population (NASH grade, cirrhosis). In cirrhosis both sensitivity and specificity of TE is 80-90% for example. So in the end we decided to refrain from going into a detailed discussion on these promising techniques (in addition to not adding additional length to this already long paper). We hope this is acceptable to the reviewer.
4. page 4. NASH diagnosis currently requires biopsy verification by histopathology, by assessing e.g. collagen content, fat content, inflammation and hepatocyte damage (Figure 2). This reviewer believes that the authors did not go into detail regarding the pathological diagnosis of liver biopsy for NAFLD. Importantly, immunostaining for PDGFRB is not common; Azan staining and Masson-trichrome staining are common. Please cite the important review on pathological diagnosis of NAFLD
Authors response: PDGFRB staining removed and review cited
5. The authors discuss advances in imaging on a cell-by-cell basis, but they should discuss which cell imaging is closest to clinical application when categorized into (1) the cellular level, (2) animal models, and (3) in humans.
Authors response: We have added some additional summary sections to each chapter and chapter 5, to guide the reader to which tracers are closest to the clinic and which are most promising.
Round 2
Reviewer 2 Report
This paper will be accepted by the International Journal of Molecular Sciences.